# DISENTANGLING STYLE AND CONTENT FOR LOW RESOURCE VIDEO DOMAIN ADAPTATION: A CASE STUDY ON KEYSTROKE INFERENCE ATTACKS

## ABSTRACT

Keystroke inference attacks are a form of side-channels attacks in which an attacker leverages various techniques to recover a user's keystrokes as she inputs information into some display (for example, while sending a text message or entering her pin). Typically, these attacks leverage machine learning approaches, but assessing the realism of the threat space has lagged behind the pace of machine learning advancements, due in-part, to the challenges in curating large real-life datasets. This paper aims to overcome the challenge of having limited number of real data by introducing a video domain adaptation technique that is able to leverage synthetic data through supervised disentangled learning. Specifically, for a given domain, we decompose the observed data into two factors of variation: *Style* and *Content*. Doing so provides four learned representations: real-life style, synthetic style, real-life content and synthetic content. Then, we combine them into feature representations from all combinations of style-content pairings across domains, and train a model on these combined representations to classify the content (i.e., labels) of a given datapoint in the style of another domain. We evaluate our method on real-life data using a variety of metrics to quantify the amount of information an attacker is able to recover. We show that our method prevents our model from overfitting to a small real-life training set, indicating that our method is an effective form of data augmentation.

## 1 INTRODUCTION

We are exceedingly reliant on our mobile devices in our everyday lives. Numerous activities, such as banking, communications, and information retrieval, have gone from having separate channels to collapsing into one: through our mobile phones. While this has made many of our lives more convenient, this phenomena further incentivizes attackers seeking to steal information from users. Therefore, studying different attack vectors and understanding the realistic threats that arise from attackers' abilities to recover user information is imperative to formulating defenses. The argument for studying these attacks is not a new one. A rich literature of prior works studying both attacks and defenses has assessed a wide array of potential attack vectors. The majority of these attacks utilize various machine learning algorithms to predict the user's keystrokes, (Raguram et al., 2011; Cai & Chen, 2012; Xu et al., 2013; Sun et al., 2016; Chen et al., 2018; Lim et al., 2020), but the ability to assess attackers leveraging deep learning methods has lagged due to the high costs of curating real-life datasets for this domain, and the lack of publicly available datasets.

Despite all the recent attention to keystroke inference attacks, numerous questions have gone unanswered. *Which defenses work against adversaries who leverage deep learning systems? Which defenses are easily undermined? Are there weaknesses in deep learning systems that we can use to develop better defenses to thwart state-of-the-art attacks?* These questions capture the essence of the underlying principles for research into defenses for keystroke inference atttacks. Given the back-and-forth nature of researching attacks and defenses, these questions can not be addressed because of the current inability to assess attacks with deep learning methods.

This paper aims to overcome the challenge of having limited number of labeled, real-life data by introducing a video domain adaptation technique that is able to leverage abundantly labeled synthetic

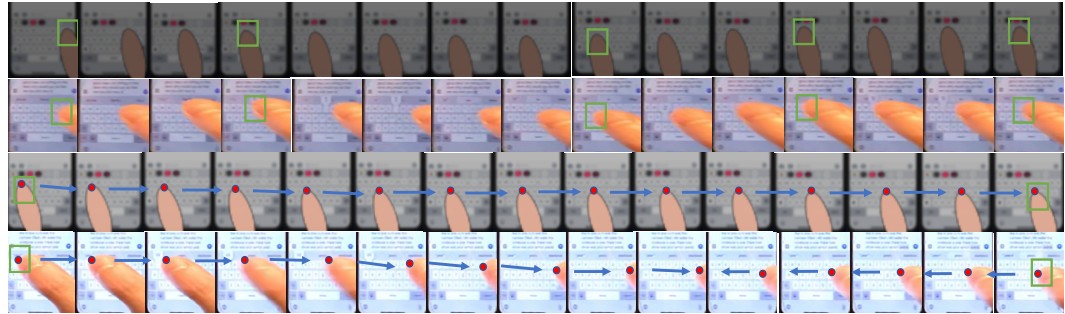

Figure 1: An example to highlight the discrepancies between the Synthetic Data (*Rows 1 and 3*) and Real-Life Data (*Rows 2 and 4*). In rows 1 and 2, we show sequences of the word *order* being typed with the same number of frames between keypresses sampled. The frames with green boxes indicate ones in which a key was pressed, i.e, in the first frame for first two rows, the key *o* was pressed. While the *content* between the two sequences is the same, the *style* is different, e.g., the texture, and trajectory in between keypresses are different. To further highlight the temporal distribution shift, we show the thumb trajectory between *w* and *h* for both synthetic and real sequences in rows 3 and 4. While the finger is linearly interpolated in the synthetic domain, the real-life one has a more complex one that is challenging to model with a simulator. We highlight the thumb tip in red and the trajectories in blue.

data. We show that by disentangling our data into separate style and content representations, we can subsequently create style-content pairs across both domains, and combine them into representations that contain the content in the style of its inputs, i.e., style transfer in the feature space. This is especially attractive in the case of pairs of real-life style and synthetic content, as this is an effective data augmentation scheme. Style representations need to be well separated between domains whereas content needs to be indistinguishable. To do this, we introduce auxiliary losses on the latent spaces to enforce disentanglement. Through a series of ablations, we show that doing so improves performance. In our context, *Content* answers the question: *What was typed?*. For example, the sentence that a user types. *Style* answers the question: *How was it typed?*. For example, the texting pattern.

The majority of visual domain adaptation methods do not work well in our problem setting because they mainly focus on tasks in which the domain shift is limited to a shift in texture, e.g., image classification, semantic segmentation, etc. (Ganin & Lempitsky, 2014; Shrivastava et al., 2016; Tzeng et al., 2017; Hoffman et al., 2017; Motiian et al., 2017). When predicting keystroke sequences, addressing the domain shift with respect to texture is not sufficient. While there is a clear difference in texture, we have to also address the temporal domain shift, e.g., different finger motions, speeds, etc. Notice the difference between the trajectories of thumbs in the two example videos displayed in Figure 1. The synthetic thumb is linearly interpolated whereas the real one moves in a more complex fashion. Our pairing mechanism is inspired by the one introduced by Motiian et al. (2017). They devise a training regime that pairs the scarce data in the target domain with the data from the source domain. This strategy aims to augment the data in the target domain on the order of the source domain. In our work, we loosen the restriction of needing pairs with the same label to adapt to our setting of not having paired sentences. This makes our pairing mechanism more general and applicable to other settings. To summarize, our main contributions are: **1**) A framework for low-resource video domain adaptation using supervised disentangled learning. **2**) A novel method to assess the threat of keystroke inference attacks by an attacker using a deep learning system while having limited real-life data.

## 2 BACKGROUND

**Keystroke Inference Attacks** Some of the early works in (vision-based) keystroke inference attacks have focused on direct line of sight and reflective surfaces *(i.e., teapots, sunglasses, eyes)* (Backes et al., 2008; 2009; Raguram et al., 2011; Xu et al., 2013; Yue et al., 2014; Ye et al., 2017; Lim et al., 2020) to infer sensitive data. The attackers train models that account for various capture angles by aligning the user's mobile phone to a template keyboard. Collectively, these works showed that

attackers are able to successfully recover pins and full sentences. In this work, we advance the state-of-the-art under the direct line of sight model wherein the attacker uses a mobile camera to record a victim's mobile phone usage. None of these works adequately explore the capabilities of an attacker that leverages deep learning systems because of the costs to collect large scale datasets. Lim et al. (2020) created a simulator that generates synthetic data for keystroke inference attacks and showed that training with both synthetic and real data, in a supervised domain adaptation framework, yielded a CNN that generalized to a real-life test set, despite having limited labels in the real domain. This work is limited due to the restricted threat scenario of inferring single keypresses. In our work, we assess the ability of an attacker to recover complete sequences. Predicting entire sequences from an input video is not only a more challenging task, but also it is a more realistic threat scenario.

**Style and Content Disentanglement in Videos** Tenenbaum & Freeman (1997); Tenenbaum & Freeman (2000) observe that by learning to factor observations of data into two independent factors of variation, *style* and *content*, models learn separate representations that can extrapolate style into novel content, classify content in different styles, and translate new content into new styles. This framework has been extended to videos and has been explored in a variety of settings. Prior works have disentangled videos into a time-dependent *style* representation and time-independent *content* with adversarial training (Denton & Birodkar, 2017; Villegas et al., 2017) or with variational autoencoders (Li & Mandt, 2018; Hsieh et al., 2018). In our setting, style and content are both time-dependent. Style encapsulates the trajectory of the finger in between keys or speed of the user typing. The difference in texture on a per-frame basis is also encapsulated by style. Content represents the entire trajectory as that determines the sentence that was typed. These methods are all unsupervised methods to disentangle style and content. Since we have labels, we are able to leverage the observation made by Locatello et al. (2018; 2019), arguing that learning disentangled representations is impossible without supervision, and that the unsupervised methods leveraging temporal inductive biases do not lead to improved disentangled representations.

**Low Resource Domain Adaptation** We are operating in a low resource setting in which we have abundant labels in the source domain and have very few, albeit labeled, data points in the target domain. Hosseini-Asl et al. (2019) extend the CyCada (Hoffman et al., 2017) and CycleGAN (Zhu et al., 2017) frameworks to the low resource domain adaptation setting by adding a semantic consistency loss. Motiian et al. (2017) addresses this problem by learning a feature space that is domain invariant, but is semantically aligned across both domains by introducing a pairing process that pairs feature samples in the training set into four groups: 1) both samples from domain A with same labels; 2) a sample from each domain with same labels; 3) both samples from domain A with different labels; 4) a sample from each domain with different labels. They use adversarial training to learn a feature representation such that a discriminator can't distinguish samples from groups 1 and 2, and also from groups 3 and 4. We extend this pairing mechanism by relaxing the constraint of needing the same labels in both domains, i.e., pairs of synthetic and real sentences. Since we are effectively transferring different styles onto the content latent space, we do not need labels in the target domain so long as they are effectively disentangled.

## 3 METHODS

We first give a brief introduction to keystroke inference attacks and define the problem setup. Then, we describe our proposed framework to disentangle the style and content latent spaces to train on all style-content pairs An overview of our method is in Figure 2 and in Algorithm 1.

### 3.1 KEYSTROKE INFERENCE ATTACKS

We model the keystroke inference attack as a Seq2Seq (Sutskever et al., 2014) problem where the input $X = \{x_1, x_2, ..., x_k\}$ is a video with $k$ frames and $Y = \{y_1, y_2, ..., y_j\}$ is a sequence of $j$ characters. The videos are of users typing on their mobile phones that are cropped and aligned to a template image. The tokens are a sequence of characters of the sentence the user typed. We do not use any paired data (i.e. the synthetic and real-life datasets do not contain the same sentences), and do not have access to any auxiliary labels such as the exact frame in which a key was pressed. Our goal is to learn the parameters of a model that maximizes the conditional probability of $Y$ given $X$. We use a Transformer (Vaswani et al., 2017) encoder-decoder as our model. In our setting we have a dataset of synthetic videos, $\mathcal{D}_s = \{(X_i^s, Y_i^s)\}$, and a dataset of real-life videos $\mathcal{D}_t = \{(X_i^t, Y_i^t)\}$,

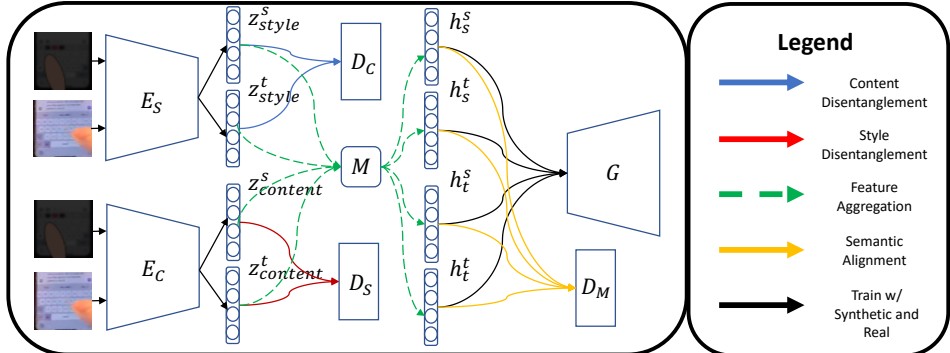

Figure 2: An overview of our training procedure. A single training iteration takes a pair of synthetic and real-life videos. We disentangle them into style and content representations, and create four combinations of feature representations. For example, real style paired with synthetic content. Style disentanglement, shown in Red, removes style information from the content space. Content disentanglement, shown in Blue, removes content information from the style space. The Green paths indicate the different ways in which we can combine the style and content representations from the two domains. Finally, we further apply a semantic alignment discriminator to the combined space, shown in Yellow, to ensure the content remains constant, regardless of style. (Best viewed in color)

where the number of real-life videos is significantly less than the synthetic (Figure 3). While a large synthetic dataset can be easily generated, there exists a distribution shift between the two domains (Figure 1). When the amount of labeled data is scarce, it becomes difficult to train neural networks that generalize to samples out of the training set.

## 3.2 DISENTANGLING STYLE AND CONTENT

Our method to address the lack of real-life data is to train on combinations of style and content representation pairs from the synthetic and real domains. We introduce auxiliary losses to enforce disentanglement of style and content, ensuring that the style latent space does not contain any information about the content, and vice versa. Our training framework consists of a Content Encoder, $E_C$, a Style Encoder $E_S$, a Decoder $G$, a Feature Aggregation Module, $M$, a Style Discriminator $D_S$, a Content Discriminator $D_C$, and a Domain-Class Discriminator $D_M$.

**Pretraining Synthetic Model** We first pretrain an Encoder-Decoder Transformer on only synthetic data. We train this network with a multi-class cross entropy loss where the goal is to predict the correct sentence for a given video. Then $E_C$, $E_S$, and $D_C$ are initialized with the weights of the pretrained Encoder, and $G$ is initialized with the weights of the pretrained Decoder.

**Style Disentanglement** Style disentanglement ensures that style information is removed from the content latent space. The content latent space is defined as $z_{content}^f = E_C(X_i^f; \theta_{E_C})$ where $f \in \{s, t\}$ where $f^s$ and $f^t$ represent the synthetic and real domains, respectively. Similar to the setup of GANs (Goodfellow et al., 2014) the Style Discriminator, $D_S$ is trained to classify whether $z_{content}^f$ is real or synthetic. Next, $E_C$ is trained to spoof $D_S$ and generate a content feature representation that is domain invariant. $D_S$ is trained using Equation 1. $E_C$ is trained using the same equation, but the labels are flipped and $D_S$ is not updated.

$$\mathcal{L}_{Adv_{D_S}} = -\mathbb{E}[\log(D_S(E_C(X_i^s))) - \log(1 - D_S(E_C(X_i^t)))] \tag{1}$$

**Content Disentanglement** Content disentanglement ensures that content information is removed from the style latent space. The style latent space is defined as $z_{style}^f = E_S(X_i^f; \theta_{E_S})$ where $f \in \{s, t\}$. The Content Discriminator, $D_C$, is a Transformer Decoder, and is trained to predict the correct sentence given the input style representation. $E_S$ is trained to spoof $D_C$ and generate a style feature representation, $z_{style}^f$, such that $D_C$ can not predict the correct sentence. This is done by maximizing the entropy, $H$, of the predictions of $D_C$. $D_C$ is trained by minimizing Equation 2. $E_S$ is trained by maximizing Equation 3 with the weights of $D_C$ kept frozen.

$$\mathcal{L}_{Adv_{D_C}} = -\log p(Y_i^z | D_C(E_S(X_i^f))) \tag{2}$$

$$\mathcal{L}_{Adv_{E_S}} = H(Y_i^z | D_C(E_S(X_i^s))) \tag{3}$$

**Feature Aggregation** A Feature Aggregation Module, $M$, combines the disentangled representations from the previous two steps. For any given pair of style and content representations we have:

$$M(z_{style}^f, z_{content}^{f\prime}) = \mathbf{m}(z_{style}^f + z_{content}^{f\prime}) \tag{4}$$

In Equation 4, $\mathbf{m}$ is the LayerNorm operation (Ba et al., 2016), $f \in \{s, t\}$ and $f' \in \{s, t\}$. There are four different possible pairs that can be the input to our model, since there are two factors of variation (*style* and *content*) and two domains (synthetic and real-life). For any given input pair, the output feature representation of $M$ can be thought as the *content* in the *style* of the specified domain. We denote this as $h_{f\prime}^f$, where $f$ is the style and $f\prime$ is the content.

**Prediction** The Decoder, $G$, takes in the output of $M$, $h_{f\prime}^f$, and outputs the predicted sentence $\hat{Y}^{f\prime}$, and is trained with the cross-entropy loss using the labels $Y^{f\prime}$. The objective is:

$$\mathcal{L}_{cls} = -\log p(Y^{f\prime} | G(M(E_S(X^f), E_C(X^{f\prime})))) \tag{5}$$

At test time, the model outputs the most likely sentence given a real-life video:

$$\arg\max_Y p(Y^t | G(h_t^t)) \tag{6}$$

**Semantic Aligment** We extend the framework of Motiian et al. (2017) to create training pairs to compensate for limited data in one domain. Rather than train on pairs of feature samples, we train on the outputs of a feature aggregation module that takes in as input style-content pairs. Furthermore, we do not need to have the same labels for both domains, i.e., we do not need to have the same synthetic and real sentences. We create four pairs $\mathcal{G}_k, k \in \{1, 2, 3, 4\}$. $\mathcal{G}_1$ and $\mathcal{G}_2$ are outputs of $M$ that share synthetic content: (Synthetic Style, Synthetic Content) and (Real Style, Synthetic Content). $\mathcal{G}_3$ and $\mathcal{G}_4$ share real content: (Synthetic Style, Real Content) and (Real Style, Real Content). A multi-class discriminator, $D_M$, is trained using Equation 7 to correctly identify which group every output of $M$ belongs to. $l_k$ is the corresponding label for a given $\mathcal{G}_k$. $E_C$, $E_S$, and $M$ are updated with Equation 8 such that $D_M$ can't distinguish outputs of $M$ that are in $\mathcal{G}_1$ and $\mathcal{G}_2$ and outputs of $M$ that are in $\mathcal{G}_3$ and $\mathcal{G}_4$.

$$\mathcal{L}_{Adv_{D_M}} = -\mathbb{E}[\sum_{k=1}^4 l_k \log(D_M(M(\mathcal{G}_k)))] \tag{7}$$

$$\mathcal{L}_{Adv_M} = -\mathbb{E}[l_1 \log(D_M(M(\mathcal{G}_2))) - l_3 \log(D_M(M(\mathcal{G}_4)))] \tag{8}$$

The final loss function to train our model is show in Equation 9 where the weightings for each term are tuned using a validation set. An overview of the training procedure is shown in Algorithm A.9.

$$\mathcal{L} = \lambda_1 \mathcal{L}_{cls} + \lambda_2 \mathcal{L}_{Adv_M} + \lambda_3 \mathcal{L}_{Adv_{D_M}} + \lambda_4 \mathcal{L}_{Adv_{E_S}} + \lambda_5 \mathcal{L}_{Adv_{D_C}} + \lambda_6 \mathcal{L}_{Adv_{D_S}} \tag{9}$$

## 4 EXPERIMENTS

In this section, we detail the datasets, the motivation and interpretation of our evaluation metrics, and experimental results. Further details regarding data collection, network architectures, and training details are located in section Appendix A.

**Datasets** Figure 3 shows different statistics for the synthetic and real datasets. We set aside 10% of the training set as a validation set. The real-life dataset was collected by recording participants typing sentences into a mobile phone. Three different participants were asked to type conversational text messages into their mobile devices while we recorded them in both indoor and outdoor settings. We used a mobile camera and captured from a distance of 3 meters. The details to preprocess and align the real data is detailed in A.2. The synthetic data was generated using the simulator by Lim et al. (2020). It generates aligned videos of a synthetic thumb typing various sentences. We generated sentences from the "A Million News Headlines" dataset released by Kulkarni (2018). We add a START and STOP token to the beginning and end of a sentence, respectively. In total, there are 30 tokens in which the decoder can predict: 26 letters and 4 special tokens (START, STOP, SPACE, PAD).

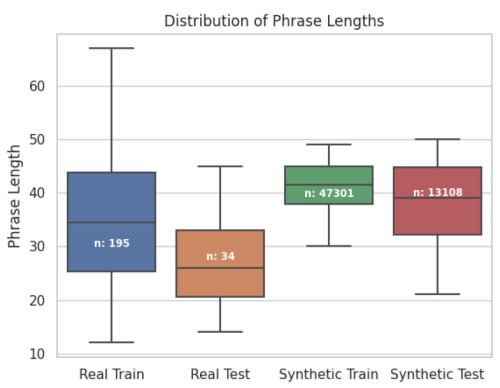 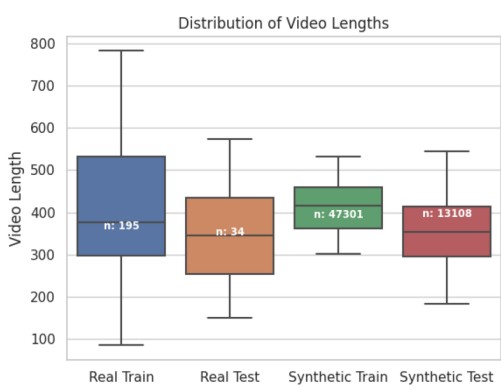

Figure 3: Distribution of phrase and video lengths for our datasets. The number of real-life data-points (229) is significantly less than synthetic (60,409).

**Evaluation Metrics** We propose to use a variety of metrics to quantify the amount of information the attacker is able to recover from the user as there is no single, agreed-upon metric for keystroke inference attacks. In the first scenario, we postprocess the outputs of our model with a language model, similar to the methodologies of Raguram et al. (2011); Xu et al. (2013); Sun et al. (2016); Chen et al. (2018). Appropriate metrics for this scenario are Bleu-n (Papineni et al., 2002), ROUGE (Lin, 2004), and METEOR (Banerjee & Lavie, 2005). Bleu-n scores are scored on n-gram precision, i.e., the n-grams in the predicted sentence that are also in the ground truth sentence. For brevity, we do not report Bleu-2 and Bleu-3. ROUGE scores are scored on n-gram recall, i.e., the n-grams in the ground truth that are also in the predicted sentence. METEOR is a metric that is scored on the harmonic mean of unigram precision and recall and was developed to address some of the drawbacks of ROUGE and Bleu-n. METEOR scores range from 0 to 1. Scores above 0.5 reflect understandable translations and scores above 0.7 reflect fluent ones (Lavie, 2010).

While these scores have merit in the context of keystroke inference attacks, they are not without shortcomings. These scores are especially harsh for predictions that contain slight typographical errors (e.g., "hello" vs. "hellp"), and there is no guarantee that the previously mentioned postprocessing steps will address every error. Also, there are settings in which the applicability of these metrics does not make sense — e.g., recovering alphanumeric passwords. Thus, we also need evaluation metrics for the raw outputs of our model. Two appropriate metrics are Translation Edit Rate (TER) (Snover et al., 2006) and a QWERTY-keyboard-based edit distance. Both metrics measure the number of edits required for a hypothesis sentence to be translated to the ground truth. The latter is a form of the Damerau–Levenshtein (DL) distance (Damerau, 1964) that penalizes the edit operations (i,e., insertions, deletions, substitutions, character swapping) conditioned on the QWERTY keyboard layout. For example, if "hello" was the ground truth word, "hellp" should be less penalized than "hellv" as the former is a more likely output than the latter given the assumed keyboard layout.

**Synthetic Data Experiments** Table 1 shows the results for a model trained and tested on synthetic data. The model performs very well on the synthetic test set across all proposed evaluation metrics. To lessen the compute cost of processing over 45k raw videos, we extract a fixed 128-dimensional feature representation as a preprocessing step by training a CNN for single key press classification. We use the simulator to generate single key press images and train a CNN to predict the correct key. Further details for this step are available in A.4.

| Method | Bleu-1 ↑ | Bleu-4 ↑ | METEOR ↑ | ROUGE ↑ | TER ↓ | Qwerty-D ↓ |
|---|---|---|---|---|---|---|
| Synthetic | 0.90 | 0.79 | .9 | 0.91 | 0.03 | 1.87 |
| Finetuning | 0.15 | 0 | 0.06 | 0.13 | 0.81 | 45.6 |
| ADDA (Tzeng et al., 2017) | 0.15 | 0 | 0.07 | 0.16 | 0.78 | 46.1 |
| CycleGAN (Zhu et al., 2017) | 0.17 | 0 | 0.07 | 0.17 | 0.7 | 45.6 |
| Ours (Base) | 0.73 | 0.52 | 0.74 | 0.75 | 0.12 | 7.4 |
| Ours (No LM) | 0.78 | 0.57 | 0.75 | 0.76 | **0.09** | **5.3** |
| Ours | **0.81** | **0.62** | **0.8** | **0.81** | **0.09** | **5.3** |

Table 1: We report various metrics to quantify the attacker's ability to recover information.

## 4.1 BASELINE RESULTS

We compare our work against finetuning, Adversarial Discriminative Domain Adaptation, ADDA, (Tzeng et al., 2017) and CycleGAN (Zhu et al., 2017). All methods are evaluated on the real-life test set and use the model trained on synthetic data.

- **Finetuning**. We finetune a model trained only on synthetic with the real-life training set.
- **ADDA**. Our goal with ADDA is to have the Encoder output feature representations that are domain invariant to a Discriminator, but also be discriminative for the Decoder.
- **CycleGAN**. This method learns a pixel-wise transformation that transforms data from one domain to another. We apply this transformation to every real frame to a synthetic one. Then, we finetune the synthetic model with the transformed real training set and test on the transformed real test set. Finetuning is needed because this method does not address the temporal shift as the transformations are conducted on a per-frame basis.

It is important to note that in Lim et al. (2020), the authors were successful in applying ADDA to the task of single key press classification when the number of labeled data is scarce. Simply applying ADDA to our sequence prediction task leads to severe overfitting due to the limited real-life data, indicating that this task is more challenging than the single keypress classification task studied by Lim et al. (2020). While these are common approaches used in domain adaptation, we found that these approaches are not suitable in our problem setting. We carried out extensive experiments to tune our baselines and maximize their performance; a full listing of these experiments and corresponding hyperparameters is available in Appendix A. Despite an extensive search of hyperparameters, we still overfit. We report the best results in Table 1.

## 4.2 ADAPTING TO REAL-LIFE VIDEOS

Our method, unlike the above baselines, does not overfit to the real-life training set. Our results show that training with our pairing mechanism with disentangled representations across domains is an effective form of data augmentation. We outperform the baselines in both raw output evaluations and post-processed evaluations as shown in Table 1. We found that our training was not sensitive to the hyperparameters and weightings of the loss terms in 9, and use the same hyperparameters for all experiments. Additional training details are in A.12.

While a direct comparison to the state of the art in direct line of sight attacks is difficult due to the differences in datasets, it is worth noting how our model performs relative to current methods. Raguram et al. (2011) achieve a METEOR score of 0.89 whereas Xu et al. (2013) achieve a score of 0.71, but recording from much farther distances. To measure an attacker's ability to recover passwords, Raguram et al. (2011) report precision and recall for individual word units and characters. They achieve word-level precision and recall of 75% and 78%, respectively, and character-level scores of 94% and 98%. We achieve a word-level precision and recall of 78% and 79%, respectively, and a precision and recall of 96% and 95%, respectively, for characters. They do not report METEOR scores for this scenario; we do so in Table 1 under "No LM"

| Method | Bleu-1 ↑ | Bleu-4 ↑ | METEOR ↑ | ROUGE ↑ | TER ↓ | Qwerty-D ↓ |
|---|---|---|---|---|---|---|
| **I -** Base | 0.73 | 0.52 | 0.74 | 0.75 | 0.12 | 7.4 |
| **II -** Base + Style | 0.77 | 0.56 | 0.76 | 0.77 | 0.1 | 6.3 |
| **III -** Base + Content | 0.77 | 0.53 | 0.76 | 0.78 | 0.11 | 5.9 |
| **IV -** Base + Style + Content | 0.76 | 0.57 | 0.75 | 0.76 | 0.12 | 7.0 |
| **V -** Base + (Motiian et al., 2017) | 0.77 | 0.57 | 0.76 | 0.79 | 0.11 | 5.7 |
| **VI -** Full w/o LayerNorm | **0.82** | 0.61 | **0.8** | **0.83** | **0.09** | **5.2** |
| **VII -** Full | 0.81 | **0.62** | **0.8** | 0.81 | **0.09** | 5.3 |
| **I -** Base (100) | 0.58 | 0.33 | 0.56 | 0.6 | 0.2 | 13.1 |
| **II -** Base + Style (100) | 0.65 | 0.38 | 0.62 | 0.65 | 0.18 | 11.2 |
| **III -** Base + Content (100) | 0.62 | 0.37 | 0.57 | 0.61 | 0.2 | 11.3 |
| **IV -** Base + Style + Content (100) | **0.69** | 0.4 | **0.65** | **0.69** | **0.15** | **9.3** |
| **V -** Base + (Motiian et al., 2017) (100) | 0.65 | 0.34 | 0.6 | 0.65 | 0.18 | 11.3 |
| **VI -** Full w/o LayerNorm (100) | 0.67 | **0.42** | 0.64 | 0.68 | 0.18 | 10.8 |
| **VII -** Full (100) | 0.65 | **0.42** | 0.63 | 0.67 | 0.21 | 13.2 |

Table 2: We conduct ablation studies to evaluate the effectiveness of each loss component. We also evaluate performance when the number of real training videos is dropped to 100 instead of 175.

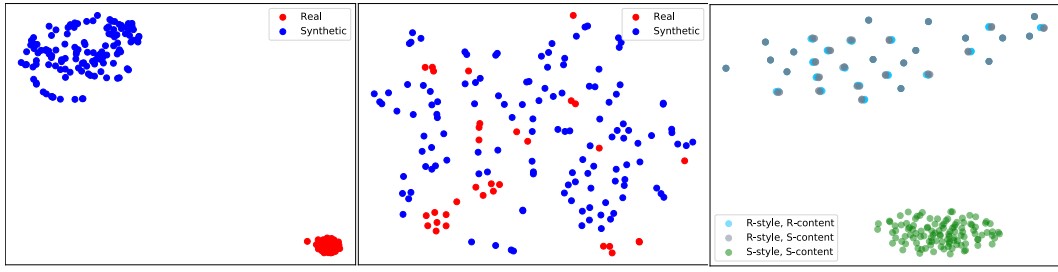

Figure 4: t-sne plots for the outputs of $E_S$ (Left), $E_C$ (Center), and $M$ (Right).

**Feature Visualization** The t-sne (Maaten & Hinton, 2008) plots, in Figure 4, for the feature representations of $E_S$, $E_C$, and $M$ on synthetic and real test data show that sentences with different styles have a noticeable separation, whereas the content representations are intertwined. The last figure on the right shows outputs of our feature aggregation module, $M$, and shows the transfer of styles in the feature space. There is a clear separation between styles, while the datapoints within one style cluster are mixed. To obtain inputs suitable for t-sne, we perform a max pooling operation along the temporal dimension of the outputs of the networks.

**Ablation** Next, we conduct a series ablation studies to explore the effectiveness of our proposed framework. We introduce seven different models: **I** is our base method without the use of any adversarial losses, just the pairing mechanism. **II** uses style disentanglement, i.e., **I** + style disentanglement. **III** uses content disentanglement, i.e., **I** + content disentanglement. **IV** uses both style and content disentanglement. **V** is the base method with only the semantic alignment loss (Motiian et al., 2017). **VI** uses style and content disentanglement, along with the semantic alignment, but does not use LayerNorm in $M$. **VII** uses style and content disentanglement, along with the semantic alignment. This is our proposed method trained with Algorithm 1, i.e., **IV** + semantic alignment.

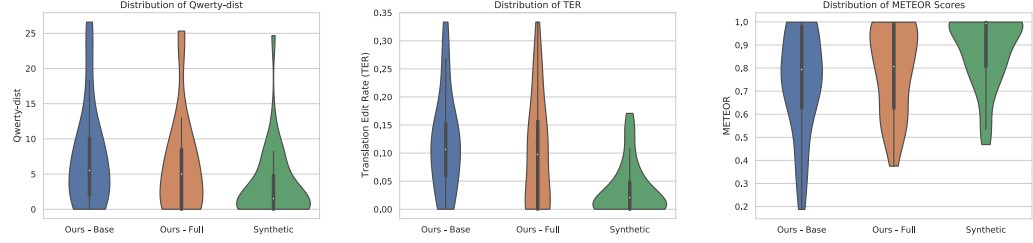

Figure 5: From left to right: the distribution of Qwerty-D, TER, and METEOR scores, respectively. "Full" is our proposed framework. "Base" is our framework without any adversarial training. We also compare these two against a model trained and tested on just synthetic data.

First, we find that our base model (**I**) achieves competitive results without any losses on the latent spaces. This indicates that training on paired representations across domains is an effective method for data augmentation. Second, we find that adding auxiliary losses on the latent spaces to enforce style and content disentanglement improves performance. The performance for models **II** and **III** shows the base model is benefiting from the added loss terms. The results for Model **IV** aligns with our hypothesis that explicitly disentangling style and content allows us to overcome the lack of training data in the target domain by training with all combinations of the factors of variation. Finally, we trained model **V** to apply the semantic alignment step on our paired outputs without any additional adversarial losses. This is quite competitive with **IV**, but we find the greatest performance boost when training model **VII** using both semantic alignment and disentanglement. A closer look into the distribution of the scores in Figure 5 shows that the distribution of scores for Model **VII** (Full) indicates higher overall performance compared to Model **I** (Base). Our results show that explicitly disentangling style and content by adding the adversarial losses on the latent spaces, supplements the pairing mechanism to achieve the highest performance against our evaluation metrics.

## 5 Conclusion

Our work provides the important initial step needed to formulate defenses for keystroke inference attacks in the age of deep learning. We provide the first assessment of an attacker's ability to recover sentence level information using deep learning systems, demonstrating that this attack is plausible and a significant threat to users. Such a task has been challenging due to the costs of curating a real-life dataset. We address this problem by introducing a framework for low resource video domain adaptation, that disentangles the style and content across both domains, and creates representations from all pairs of style and content combinations. Our results indicate that training with these pairs, along with auxillarly losses to explicitly disentangle style and content, serves as an effective form of data augmentation that prevents overfitting.

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

# A    ADDITIONAL TRAINING DETAILS

## A.1    SYNTHETIC DATASET COLLECTION

We use the synthetic dataset generator released by Lim et al. (2020) to generate our synthetic data. We take the headlines dataset released by Kulkarni (2018) for our sentences. To preprocess every sentence, we change it to lower case, remove any special characters, and punctuation marks. The thumb is of only right-handed users. The simulator takes in parameters for the user's device and the attacker's camera. We use the iPhone 6 as the camera parameters and the iPhone XR as the user's device.

## A.2    REAL-LIFE DATASET COLLECTION

We preprocess the real-life capture data in order to compensate for the various capturing angles and positions and adversary can take. More specifically, for every video: 1. We crop out the location of the mobile phone. 2. Manually label the 4 corners of the phone. 3. Calculate a homography $\mathbf{H}$ matrix between the labeled 4 corners and 4 corners of a template image. 4. Warp the captured image to the template image using $\mathbf{H}$. All aligned images are resized to 200 x 100 and then cropped to only display the keyboard. We capture from distances up to 3 meters where the attacker is using an iPhone6 camera and the user is using an iPhone XR.

## A.3    ALGORITHM BLOCK

---

**Algorithm 1:** Learning Algorithm for Disentangling Style and Content.

---

**Input:** $E_C, E_S, M, D_S, D_C, G, D_M, \mathcal{D}_S, \mathcal{D}_T$
**Result:** Well-trained $\hat{E}_C$, Well-trained $\hat{E}_S$, Well-trained $\hat{M}$, and Well-trained $\hat{G}$

1   **while** *Not Converged* **do**
2      sample mini-batch of $b$ synthetic samples, $\{(X_1^s, Y_1^s), \ldots, (X_b^s, Y_b^s)\}$ from $\mathcal{D}_S$
3      sample mini-batch of $b$ real-life samples, $\{(X_1^t, Y_1^t), \ldots, (X_b^t, Y_b^t)\}$ from $\mathcal{D}_T$
4      **Style Disentanglement**: Remove *Style* information from the Content Space
5      update $D_S$ and $E_C$ with $\mathcal{L}_{Adv_{D_S}}$
6      **Content Disentanglement**: Remove *Content* information from the Style Space
7      update $D_C$ with $\mathcal{L}_{Adv_{D_C}}$
8      update $E_S$ with $\mathcal{L}_{Adv_{E_S}}$
9      **Sequence Prediction**
10     update $E_C, E_S, G, M$ with $\mathcal{L}_{cls}$
11     **Semantic Alignment**
12     update $D_M$ with $\mathcal{L}_{Adv_{D_M}}$
13     update $M, E_C, and E_S$ with $\mathcal{L}_{Adv_M}$
14 **end**
15 **return** $\hat{E}_C = E_C, \hat{E}_S = E_S, \hat{M} = M, \hat{G} = G$

---

## A.4    SYNTHETIC SINGLE KEY PRESS CLASSIFIER

We train a CNN, $\phi(\cdot)$, for the task of single key press classification in order to learn a $d-$dimensional ($d = 128$) feature extractor. Training with almost 50k synthetic videos where the average sequence length is 387 is computationally expensive. Once this network is fully trained for the task of single key press classification, we can extract the features of each video on a per-frame basis.

**Data Preparation**    First, we use the simulator (Lim et al., 2020) to generate 70,000 single key press images. These images contain the synthetic thumb over one of 27 keys on the QWERTY keyboard (26 letters + the space bar). Once generated, these images are preprocessed in a similar fashion to the synthetic video dataset. We resize the images to size 200 X 100 and crop the phone such that only the keyboard is showing. We use 50,000 images for training and 10,000 images for testing and validation, respectively.

**Training Details and Results**  We use a CNN where each layer consists of a Convolution Layer, ReLU activation, and MaxPool operation. We use 3 layers and 2 fully connected layers. We were able to train a network to achieve 95% accuracy on a held out test set without much hyperparameter or architecture search, as this is a fairly simple 27-way classification task. We use this final model to preprocess every frame in our synthetic video dataset. Every video is a now a sequence of these $d-$dimensional feature representations. We use the Adam optimizer with a learning rate of $1e-3$.

## A.5 Real-Life Single Key Press Classifier

When extracting the visual features for the real-life videos, we can not use a feature extractor that was trained only on synthetic data. There is a distribution shift between the synthetic and real-life data, so the features we extract would be not be informative. Instead of using $\phi(\cdot)$ that was trained for single keypress classification on just synthetic data, we train $\phi(\cdot)$ with a combination of synthetic and real-life data. Specifically, we adopt the ADDA (Tzeng et al., 2017) framework for unsupervised domain adaptation to train $\phi(\cdot)$.

**Data Preparation**  We treat the individual frames for all of the videos in our real-life training set as unlabeled data. Even though we do not have labels for individual keypresses for real-life data, we can leverage the fact that we have abundant labels for synthetic data by adopting the unsupervised domain adaptation technique ADDA.

**Training Details and Results**  We use the CNN for single key press classification on synthetic data as our pretrained network. The Discriminator is a 1 layer, 128-dimensional fully connected layer followed by a sigmoid. We follow the same guidelines to train ADDA as the original paper (Tzeng et al., 2017), and refer the reader to this work for the full description of their training process. We use the Adam optimizer and a learning rate of $1e-3$ for both the Discriminator and CNN.

## A.6 Network Architectures

For all experiments, the Encoders ($E_C$, $E_S$) and Decoders ($D_C$, $G$) are both Transformers with 4 layers, 4 attention heads, an embedding size of 128 and a hidden size of 256. $D_M$ and $D_S$ are both 1-layer fully connected layers. Since the output of the Encoder is a sequence of $n$ continuous representations, where $n$ is the input sequence length, we do a max pooling operation along the temporal dimension so that we have a fixed vector representation. These fixed vector representations are the direct inputs to $D_M$ and $D_S$. The max sequence length is set at 300, and the max phrase length is set at 70. If an input sequence has more than 300 frames, we randomly sample 300 frames at each epoch. If a video in the testing or validation set has more than 300 frames, we fix the indices of the sampled frames to remove any randomness for evaluation. For input sequences that are shorter than 300 frames, we zero-pad the remaining sequence.

## A.7 Additional Training Details - Synthetic Seq2Seq

We use a dropout value of $p = 0.1$ and the Adam optimizer with a learning rate $1e-4$. We train the network for 400 epochs.

## A.8 Additional Training Details - Finetuning

We use the $\phi(\cdot)$ trained on a combination of real-life and synthetic data described in A.5. We do not fix extract a fixed d-dimensional representation for every frame in the real-life video in order to preprocess or reduce computational costs. Since we are dealing with such few real-life videos, handling raw videos is not as computationally expensive. Doing this lets us apply data augmentation techniques on the image space versus the feature space. Also, we can continue to adjust the weights of $\phi(\cdot)$. Table 3 shows the various configurations we experimented with for finetuning. We experimented with different combinations of freezing networks and tuning learning rates. We were unable to discover any combination that did not overfit to the training data. For all experiments, we use the Adam optimizer.

| Frozen Layers | LR-Decoder | LR-Encoder | LR-$\phi(\cdot)$ | Bleu-1 |
|---|---|---|---|---|
| None | 1e−4 | 1e−4 | 1e−4 | 0.15 |
| None | 1e−4 | 1e−5 | 1e−5 | 0.12 |
| None | 1e−5 | 1e−5 | 1e−5 | 0.12 |
| $\phi(\cdot)$ | 1e−4 | 1e−4 | 0 | 0.12 |
| $\phi(\cdot)$ | 1e−5 | 1e−5 | 0 | 0.15 |
| Encoder | 1e−4 | 0 | 1e−4 | 0.14 |
| Encoder | 1e−5 | 0 | 1e−5 | 0.14 |
| Decoder | 0 | 1e−4 | 1e−4 | 0.12 |
| Decoder | 0 | 1e−4 | 1e−5 | 0.08 |
| Encoder, Decoder | 0 | 0 | 1e−4 | 0 |

Table 3: The different hyperparameters we tuned to maximize the performance for finetuning.

### A.9 ADDITIONAL TRAINING DETAILS - ADDA

ADDA (Tzeng et al., 2017) is an unsupervised domain adaptation method that learns a feature representation that is domain invariant and discriminative. While this framework was originally proposed for UDA, it can be easily extended to a supervised domain adaptation scenario. To follow the ADDA framework, we initialize the weights of our Encoder and Decoder with the ones pretrained on synthetic data. Then, we train on mini-batches of both synthetic and real-life videos such that the Decoder can predict the input video's labels, but the Encoder produces representations that are invariant to domain, i.e., synthetic vs. real. We train in adversarial fashion where the Discriminator is trained to successfully predict the domain of its input, and the Encoder is trained to spoof the Discriminator. The Discriminator used in this experiment is the same one we use for our method, $D_S$. To get the input for $D_S$, we first perform a maxpool operation on the output of the Encoder. We also tried to use a 1 layer LSTM with hidden size of 256 as our Discriminator. For the LSTM Discriminator, the input is the entire sequence of the Transformer output — no pooling. The output for the last time step is used as input to a linear layer followed by a sigmoid for binary classification. Table 4 highlights the different hyperparameters we tried to maximize performance.

### A.10 ALGORITHM BLOCK

---

**Algorithm 2:** Learning Algorithm for ADDA.

---

**Input:** $E_C$, $G$, $D_S$, $\mathcal{D}_S$, $\mathcal{D}_T$

**Result:** Well-trained Encoder $\hat{E}_C$ and Well-trained Decoder $\hat{E}_S$

1 **while** *Not Converged* **do**

2      sample mini-batch of $b$ synthetic samples, $\{(X_1^s, Y_1^s), \ldots, (X_b^s, Y_b^s)\}$ from $\mathcal{D}_S$

3      sample mini-batch of $b$ real-life samples, $\{(X_1^t, Y_1^t), \ldots, (X_b^t, Y_b^t)\}$ from $\mathcal{D}_T$

4      **Domain Invariance**: Make Encoder outputs domain invariant

5      update $D_S$ and $E_C$ with $\mathcal{L}_{Adv_{D_S}}$

6      **Sequence Prediction**: Make Encoder outputs discriminative

7      update $E_C$ and $G$ with $\mathcal{L}_{cls}$

8 **end**

9 **return** $\hat{E}_C = E_C$, $\hat{G} = G$

---

### A.11 ADDITIONAL TRAINING DETAILS - CYCLEGAN

CycleGAN (Zhu et al., 2017) learns pixel-wise transformation functions $F : S \rightarrow T$ and $G : T \rightarrow S$ that can transform data from the source domain to the target domain, and vice versa. We take 10,000 random frames from the synthetic and real-life video datasets and train $F$ and $G$ in an unpaired fashion. We train using the default CycleGAN hyperparameters reported in Zhu et al. (2017) and train for 200 epochs. We take $G$ and transform every frame in our real-life dataset to the synthetic space. We found that images transformed under $G$ (i.e, real to synthetic) yielded higher quality transformations (e.g., the thumb was not malformed, the thumb was placed on the correct key, etc.) as opposed to the transformed images produced by $F$. Then, we finetune on the transformed real-life training set and test on the transformed real-life test set. Simply testing on the transformed

| Frozen Layers | D Architecture | LR-Decoder | LR-Encoder | LR-$\phi(\cdot)$ | Bleu-1 |
|---|---|---|---|---|---|
| None | 128-FC | 1e−4 | 1e−4 | 1e−4 | 0.13 |
| None | Lstm | 1e−4 | 1e−4 | 1e−4 | 0.15 |
| None | 128-FC | 1e−4 | 1e−5 | 1e−5 | 0.14 |
| None | Lstm | 1e−4 | 1e−5 | 1e−5 | 0.16 |
| Encoder | 128-FC | 1e−4 | 0 | 1e−4 | 0.13 |
| Encoder | Lstm | 1e−4 | 0 | 1e−4 | 0.17 |
| Decoder | 128-FC | 0 | 1e−4 | 1e−4 | 0.1 |
| Decoder | Lstm | 0 | 1e−4 | 1e−4 | 0.1 |
| Decoder | 128-FC | 0 | 1e−4 | 1e−5 | 0.08 |
| Decoder | Lstm | 0 | 1e−4 | 1e−5 | 0.07 |
| Encoder, Decoder | 128-FC | 0 | 0 | 1e−4 | 0 |
| Encoder, Decoder | Lstm | 0 | 0 | 1e−4 | 0 |

Table 4: The different hyperparameters we tuned to maximize the performance for ADDA.

real-life test set is not sufficient as the CycleGAN approach does not address temporal domain shift as the transformations are performed independently on a per-frame basis. Thus, we are finetuning to address this temporal shift. We use the $\phi(\cdot)$ from the synthetic key press classifier. Table 5 details the different configurations of learning rates and weight freezing we experimented on for finetuning. Despite finetuning on the transformed data, this method still overfits to the training set and is unable to generalize.

| Frozen Layers | LR-Decoder | LR-Encoder | LR-$\phi(\cdot)$ | Bleu-1 |
|---|---|---|---|---|
| None | 1e−4 | 1e−4 | 1e−4 | 0.17 |
| None | 1e−4 | 1e−5 | 1e−5 | 0.17 |
| None | 1e−5 | 1e−5 | 1e−5 | 0.13 |
| $\phi(\cdot)$ | 1e−4 | 1e−4 | 0 | 0.13 |
| $\phi(\cdot)$ | 1e−5 | 1e−5 | 0 | 0.15 |
| Encoder | 1e−4 | 0 | 1e−4 | 0.15 |
| Encoder | 1e−5 | 0 | 1e−5 | 0.16 |
| Decoder | 0 | 1e−4 | 1e−4 | 0.12 |
| Decoder | 0 | 1e−4 | 1e−5 | 0.14 |
| Encoder, Decoder | 0 | 0 | 1e−4 | 0.13 |

Table 5: The different hyperparameters we tuned to maximize the performance for CycleGAN.

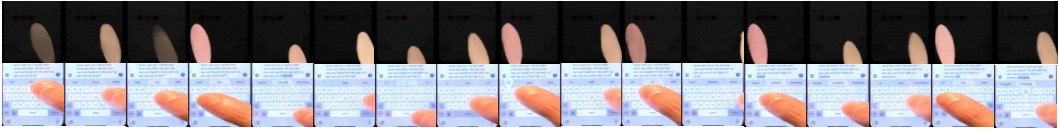

Figure 6: An example of the outputs of CycleGAN (Zhu et al., 2017). The top is the output of CycleGAN and the bottom is the original real-life image.

Figure 6 depicts an example of the outputs of the CycleGAN network we train. We show the outputs of $G : T \rightarrow S$. The top row contains the output of $G$ where the bottom row is the real-life input. The thumb for the synthetic outputs is, for the most part, on the same location as the rea-life input. While the thumb being placed on the correct location is important, it is not sufficient to fully address the domain gap between the synthetic and real. This is because this method does address the temporal shift. These transformations are done on a per-frame basis. The different thumb colors is an artifact of this method displaying temporal inconsistency.

## A.12 ADDITIONAL TRAINING DETAILS - OURS

We use the Adam optimizer with learning rate of 1e−4 for all of our networks. We train for 60k iterations with a batch size of 8, and use a dropout value of 0.15. We use the validation set to tune the

different weightings for our loss function. We set $\lambda_1 = 1.0, \lambda_2 = 0.25, \lambda_3 = 1.0, \lambda_4 = 1.0, \lambda_5 = 1.0$ and $\lambda_6 = 1.0$.

