# OpenReview forum: "Disentangling style and content for low resource video domain adaptation: a case study on keystroke inference attacks"
_ICLR.cc/2021/Conference — Reject_

### Official Review · AnonReviewer2 · 2020-10-28
**Novel with good results**

**Rating:** 7
**Confidence:** 3

**Review:**

In this paper, the authors introduce a video domain adaptation technique that learns to disentangle style and content of a video in order to generate as form of data augmentation. The main idea is to train a style and content encoder and enforce
1)	The output of the style encoder cannot be used to determine keystrokes
2)	The output distribution of the content encoder for real and synthetic data cannot be distinguished.
They motivate their problem by recognizing that deep learning-based keystroke inference attacks are trained with a small number of real data along with a larger number of synthetic data. This results in the need for data augmentation via domain adaptation.

Pros:
The proposed method of disentanglement and combination in the feature space for classification is novel and interesting. Figures are well done and informative. Multiple evaluation metrics were used, and it is clear that the framework is useful under the considered setting.


Cons/Comments:
The output of style encoder gives you a representation that maximizes the prediction loss in order to lose content information. The output of content encoder on the other hand, gives a domain invariant representations that presumably holds all the information needed for classification. The authors propose combining them with a decoder in various combination in order to do data augmentation. However, it is not clear to me how would style representation be useful in this data augmentation. In the case where style representation is totally ignored and decoder only looks at the content representation, it minimizes the prediction loss (Eq 5) and also the semantic alignment loss (Eq 8). While Table 2 shows it clearly improves the results, I am unsure of the exact reasoning on how the losses help the style encoder in encoding anything useful for the final classification.

In general, I find this paper well-written and well-motivated. The method is novel and produces good results over other baselines. I recommend this paper for acceptance.

---

> ### Author Response · Authors · 2020-11-15
> **Response to AnonReviewer2**
>
> We thank you for taking the time to read our paper and leave a review. We also thank you for the concise and informative summary of our work as it gives us confidence that we have conveyed our message well. We appreciate your thoughtful feedback to improve our paper.
>
> To address your concerns see below:
>
> -> “However, it is not clear to me how would style representation be useful in this data augmentation. In the case where style representation is totally ignored and decoder only looks at the content representation, it minimizes the prediction loss (Eq 5) and also the semantic alignment loss (Eq 8). While Table 2 shows it clearly improves the results, I am unsure of the exact reasoning on how the losses help the style encoder in encoding anything useful for the final classification.”
>
> We may be misinterpreting your question, but hopefully this clears up the potential confusion. In Table 2 we perform various ablations to see how the different loss components affect our model. When we train Model II (Base + Style), we are not making predictions only on the style information. We still use the content encoder, but we do not add any additional losses in the content latent space.
>
> We hope this addresses your concerns in a satisfactory manner. Again, we thank you for taking the time to review our work and finding our work interesting and novel.

---

> > ### Comment · AnonReviewer2 · 2020-11-20
> > **Clarification of the confusion**
> >
> > I would like to clarify my previous question. I might be overlooking some details but I am not sure intuitively, how is the model able to disentangle style and content. I would like clarification on which loss term prevents the decoder from fully ignoring the style representation. If I am not wrong, eq 5 and 8 do not prevent style representation from being ignored?

---

> > > ### Author Response · Authors · 2020-11-20
> > > **Response (Hoping to clarify confusion)**
> > >
> > > Hey,
> > > Thanks for the response.
> > > Correct, equations 5 and 8 do not ensure that style information will be ignored.
> > >
> > > When you ask how the model disentangles style and content, are you asking about the complete model? (The model with all of the loss terms)
> > >
> > > Edit: You asked for which loss term that prevents the decoder from ignoring the style representation. When would the decoder need to ignore the style representation? Are you referring to one of the ablations?
> > >
> > > Thanks!

---

### Official Review · AnonReviewer4 · 2020-10-28
**Disentangling style and content for low resource video domain adaptation: a case study on keystroke inference attacks**

**Rating:** 5
**Confidence:** 2

**Review:**

This paper is about keystroke inference attacks and proposes a method to assess the threat of deep learning based approaches
when only limited real-life data are available. To this end, it is introduced a video domain adaptation technique
that is able to generate data into separate style and content representations. This data augmentation scheme is shown to be effective and prevents overfitting.

This is an interesting applicative approach, but I believe that the introduced novelty is probably limited for ICLR. Two claimed contributions are the assessment of deep learning methods for keystroke inference attacks when limited data are available and the domain adaptation approach to generate synthetic data. Now these contributions have been also introduced recently in Lim et al. (2020). One difference is that the threat scenario is based on single keypresses,  while in this submission complete sequences are tackled. In addition, in Lim et al. it is used the adversarial discriminative domain adaptation (ADDA) technique introduced by Tzeng et al. 2017, while in this submission a supervised disentangled learning based approach is proposed.
Given these similarities I would have expected more discussion in the related work about this paper and also some comparison between single keypresses results vs complete sequences. Furthermore, from Table 1 it appears that ADDA provides much worse results with respect to the proposed method. Then a discussion is needed since it has been used in Lim et al. achieving good performance in a similar context.

For what concern the supervised disentangled learning based approach, I think that the authors should make very clear which is the introduced novelty with respect to current state-of-the-art methods. In particular, which are the new components of the proposed method. In this respect it would be helpful to make comparisons with some baselines. This would help to assess the relevance of the proposal.

As a final minor comment I think it would be important to provide more information about the real life dataset used in the experiments, for example by indicating how many participants where involved.

=============== Post rebuttal comments ===============

First of all, I want to thank the authors for answering my questions.
The clarifications confirm my previous concern about the limited technical novelty.
In addition, they highlight that only three participants were involved to build the real life dataset used in the experiments.
In my opinion this is not sufficient to carry out a significant evaluation.
For these reasons I keep my original rating.

---

> ### Author Response · Authors · 2020-11-15
> **Response to AnonReviewer4 (Part 2)**
>
> -> “it would be important to provide more information about the real life dataset used in the experiments”
>
> We have added additional details regarding the real-life dataset in the datasets subsection. Additional details regarding the real-life dataset is located in A.2.
>
>
> Again, thank you for taking the time to review our work and leaving your constructive feedback.

---

> > ### Comment · AnonReviewer4 · 2020-11-21
> > **Additional information about the dataset**
> >
> > In my review I asked to indicate how many participants where involved to build the real-life dataset collection. Unfortunately, I still cannot find this information in A.2.  I'm probably missing something, but A.2 of the revised submission is identical to the previous one. Can you check?

---

> > > ### Author Response · Authors · 2020-11-22
> > > **Clarification for information regarding the dataset**
> > >
> > > We added this to "Datasets" paragraph in the Experiments section: "Three different participants were asked to type conversational text messages into their mobile devices while we recorded them in both indoor and outdoor settings."
> > > Apologies for the confusion!

---

> ### Author Response · Authors · 2020-11-15
> **Response to AnonReviewer4 (Part 1)**
>
> Thank you for taking the time to read and review our work. We appreciate the concise summary and constructive feedback. We hope to address your main concerns regarding the novelty of our work in a satisfactory manner.
>
> -> “This is an interesting applicative approach, but I believe that the introduced novelty is probably limited for ICLR. Two claimed contributions are the assessment of deep learning methods for keystroke inference attacks when limited data are available and the domain adaptation approach to generate synthetic data. Now these contributions have been also introduced recently in Lim et al. (2020).”
>
> One of the main concerns for the reviewer is regarding the novelty of our two proposed contributions: 1) “the assessment of deep learning methods for keystroke inference attacks when limited data are available” and 2) “the domain adaptation approach”. We address our novelty with respect to the state of the art in the assessment of deep learning methods for keystroke inference attacks when limited data are available (i.e., Our proposed contribution 1) While Lim et al. (2020) have also assessed deep learning methods for keystroke inference attacks when real-life data is limited, their study is focused on single keypresses. For the task of predicting a single keypress, the input is a single frame and the output is a prediction over a qwerty keyboard (n-way classification task). In our work, we focus on predicting sequences where the input is a video of a user typing on their phone, and the output is the predicted sequence (for example, a sentence). We model this task a seq2seq problem and is a much more challenging scenario which requires a more advanced domain adaptation method. We added additional text in the "Related Works" section and the "Baseline Results" subsection to highlight the difference between our work and Lim et. al 2020
>
> -> “For what concern the supervised disentangled learning based approach, I think that the authors should make very clear which is the introduced novelty with respect to current state-of-the-art methods.”
>
> One of the main concerns for the reviewer is regarding the novelty of our two proposed contributions: 1) “the assessment of deep learning methods for keystroke inference attacks when limited data are available” and 2) “the domain adaptation approach”. We address our novelty with respect to the state of the art in the domain adaptation approach (i.e., Our proposed contribution 2)
>
> The main novelty with respect to the current state-of-the-art methods for domain adaptation is the pairing strategy we use during training, and the general framework for low resource video domain adaptation, a problem that is not well-researched. Recall that, given input data from the real and synthetic domains, we decompose the data into style and content representations for both domains, leaving us with real-style, real-content, synthetic-style, and synthetic-content. Next, we form all possible style and content pairs, and aggregate the style and content features into one representation, i.e., (real-style, synthetic-content). This pairing mechanism relaxes the constraints on the pairing strategy introduced by FADA (Moiitan Neurips 2017) for few shot domain adaptation. FADA requires one to have the same labels in both the source and target. Our method does not require one to have the same labels (sentences) across domains. We added additional text in the introduction to further highlight this.
>
> -> “ Given these similarities I would have expected more discussion in the related work about this paper and also some comparison between single keypresses results vs complete sequences. Furthermore, from Table 1 it appears that ADDA provides much worse results with respect to the proposed method. Then a discussion is needed since it has been used in Lim et al. achieving good performance in a similar context.”
>
> This is a great point as our original draft misses further discussion distinguishing ourselves from Lim et. al 2020. We have added additional discussion in the related works to further highlight the differences between our work (predicting sequences) and the work of Lim et. al (predicting single keypresses). We have added additional discussion in the results/experiments section discussing why we believe the ADDA method performs so poorly in our method, but excels for single keypresses (Lim et. al 2020). Furthermore, we have added additional training details + an algorithm codeblock in the appendix (A.10) to showcase how we applied ADDA for a seq2seq problem. As mentioned earlier, ADDA does not work well for our problem as the task of sentence prediction is much more challenging and our model overfits due to the limited number of real-life data.

---

> > ### Comment · AnonReviewer4 · 2020-11-21
> > **Further discussion on the technical novelty**
> >
> > I want to thank the authors for their discussion about the introduced contributions and for improving the submission so as to better clarify the novelty with respect to (Lim et al. 2020). The authors claim the following two contributions:
> > 1) “the assessment of deep learning methods for keystroke inference attacks when limited data are available” and
> > 2) “the domain adaptation approach”.
> >
> > For what concern the proposed domain adaptation approach if I understand well the main novelty is the pairing strategy used during training if compared with (Moiitan et al. 2017).
> > In this respect I believe that the innovation is a bit limited and this is also the reason why I asked for additional comparisons using more baselines (literature on few shot learning and disentangled representations is very extensive).
> > However, if I underestimated the technical novelty, please add more discussion on this point.

---

> > > ### Author Response · Authors · 2020-11-22
> > > **Response regarding technical novelty**
> > >
> > > Thank you for the response!
> > >
> > > Thanks for the taking the time to better understand our paper. We greatly appreciate this.
> > >
> > > Yes, the main technical novelty is the pairing strategy as it doesn't require one to have the same labels in both domains as in (Moiitan et al. 2017).
> > >
> > > There weren't too many few shot learning methods that we believed could easily work for our problem. What baselines did you have in mind? In regards to disentangled learning baselines, did you mean baselines to better disentangle style and content? We did not think this was necessary as domain confusion and entropy maximization are fairly common methods to do disentangle style and content:
> > >
> > > (Peng et al 2019) http://proceedings.mlr.press/v97/peng19b/peng19b.pdf.
> > >
> > > (John et al 2018) https://arxiv.org/pdf/1808.04339.pdf
> > >
> > > (Fu et al 2018) https://arxiv.org/pdf/1711.06861.pdf.
> > >
> > > (Springenberg et al 2016) https://arxiv.org/pdf/1511.06390.pdf.

---

### Official Review · AnonReviewer3 · 2020-10-29
**Lacks of novelty and some experiment results**

**Rating:** 5
**Confidence:** 4

**Review:**

The authors investigate video domain adaption for keystroke inference attacks on synthetic and real-life data. They propose a disentangle model to separate the representation of content and style. In their problem, style encapsulates the trajectory of the finger in between keys or speed of the user typing, while content represents the typed sentence. The model is naive by simply applying the adversarial domain adaptation into the keystroke inference attacks. Also, I suspect the paper is more suitable for the security conference, e.g., S&P. In addition, there still exists the following concerns.

+ lack comparison to the state-of-the-art methods, such as:
   John Lim et al, 'Revisiting the Threat Space for Vision-based Keystroke Inference Attacks', in arXiv preprint arXiv:2009.05796, 2020.

+ lacks comparison to the state-of-the-art domain adaptation methods, such as:
    Ehsan Hosseini-Asl et al, 'AUGMENTED CYCLIC ADVERSARIAL LEARNING FOR LOW RESOURCE DOMAIN ADAPTATION', in ICLR 2019.

+ In addition to CycleGAN, video to video translation methods should be included:
  Ting-Chun Wang et al, 'Video-to-Video Synthesis', in NeurIPS 2018.

+ I notice that the generated results quality of CycleGAN (in Figure 6) is poor. The background is black and only a thumb is generated. Please explain the reason.

+ The paper claims that the proposed method can separate the style () and content. The visualization or demonstration is needed to support their claims.

---

> ### Author Response · Authors · 2020-11-15
> **Response to AnonReviewer3 (Part 3)**
>
>
> -> “In addition to CycleGAN, video to video translation methods should be included: Ting-Chun Wang et al, 'Video-to-Video Synthesis', in NeurIPS 2018.”
>
> This is a very good point and we are glad you brought this up! We have spent significant time applying vid2vid in our scenario, but struggled to generate output videos with plausible thumb trajectories.
>
> Here is the setup we used to train vid2vid:
>     1) Generate synthetic versions of the real-life training data
>     2) We need to trim the videos such that the lengths of the videos are the same.
>     3) Train the vid2vid setup using the official github repo (https://github.com/NVIDIA/vid2vid) using the pairs of synthetic and real videos
>     4) Run the vid2vid test script on the real-life test videos to transform real videos to synthetic
>     5) Either transform the real-life training videos to synthetic and finetune on these videos OR directly test on the transformed real-life test videos
>
> We believe we have issues in generating plausible videos from vid2vid because we do not have the level of supervision that this method assumes.  Vid2Vid (https://arxiv.org/abs/1808.06601) and Few shot Vid2Vid (https://arxiv.org/abs/1910.12713)  are supervised methods, requiring paired examples. The datasets used in vid2vid all have supervision on a per-frame basis. Cityscapes and Apolloscape are street scene videos where one has access to real-life video foottage along with the semantic labels. The face video dataset has pairs of real-life talking faces alongside facial landmark labels. The dance video dataset has pairs of real-life dancing videos alongside pose labels generated from OpenPose and DensePose. For a given pair of real and synthetic videos,  we have global supervision (i.e., the same sentences), but we do not have “local” supervision (i.e. the frames are not paired) because the synthetic and real thumbs have different temporal dynamics. A training pair for Cityscapes has the same temporal dynamics because one domain is the real-life videos and the other domain is that frame's semantic labels.
>
> The similarity between our keystroke dataset setup and the authors of vid2vid is that the source and target videos are semantically similar in a global setting (The entire videos are of the same thing). The difference between our keystroke dataset setup and the authors of vid2vid is that the individual frames for the source and target videos are not the same whereas the individual frames for the source and target videos is the same for the authors of vid2vid. For example, any given frame of the cityscapes dataset is always paired with the semantic segmentation labels for the frame. In the dancing videos dataset, any given frame of a human dancing is always paired with the pose labels. In the context of keystroke videos, the thumbs would need to be in the same place. We believe that vid2vid only works when the temporal dynamics between the source and target domains are exactly the same, which is not true for the case of pairs of synthetic and real keystroke videos.
>
>
> -> “I notice that the generated results quality of CycleGAN (in Figure 6) is poor. The background is black and only a thumb is generated. Please explain the reason.”
>
> We happened to choose an example that has a very dark background which may and will confuse the reader. In this example, the background is not black ---just very dark. The background is so dark because some of the videos in our synthetic data have very dark backgrounds. We use the simulator introduced by (Lim et. al 2020) to generate synthetic videos and the darkness varies. We will find another sample in our generated outputs that do not have such a dark background for the camera ready version. Thanks for pointing this out!
>
> -> “The paper claims that the proposed method can separate the style () and content. The visualization or demonstration is needed to support their claims.”
>
> Please take a look at Figure 4 as such a visualization exists there.
>
>
> We thank you for your detailed review and feedback. We hope we addressed all of your concerns in a satisfactory manner.

---

> ### Author Response · Authors · 2020-11-15
> **Response to AnonReviewer3 (Part 2)**
>
> -> “lack comparison to the state-of-the-art methods, such as: John Lim et al, 'Revisiting the Threat Space for Vision-based Keystroke Inference Attacks', in arXiv preprint arXiv:2009.05796, 2020.”
>
> This is a great point and are glad that you brought this up. The original version lacked additional discussion situating our work with respect to Lim et. al 2020. The work in Lim et. al 2020 focuses on predicting single keypresses (an n-way classification task) while our work focuses on predicting entire sentences from videos (a seq2seq problem). In Lim et. al 2020, they used Adversarial Discriminative Domain Adaptation (Tzeng et al 2017) as their domain adaptation method. ADDA proved to be highly effective when dealing with the task of single keypress classification. Lim et. al 2020 are able to successfully train a single keypress classifier with very few real-life samples. We attempt to use ADDA for our task of predicting sequences, but, as we show in Table 1, ADDA leads to severe overfitting as our task is a much more complex task compared to single keypress classification. We have added additional text in the manuscript to further contextualize our work with respect to Lim et. al 2020. Specifically:
>     1) We have added additional discussion in the related works to further highlight the differences between our work (predicting sequences) and the work of Lim et. al (predicting single keypresses)
>     2) We have added additional discussion in the results/experiments section discussing why we believe the ADDA method performs so poorly in our method, but excels for single keypresses (Lim et. al 2020). Furthermore, we have added additional training details + an algorithm codeblock in the appendix (A.10) to showcase how we applied ADDA to our problem.
>
> -> “lacks comparison to the state-of-the-art domain adaptation methods, such as: Ehsan Hosseini-Asl et al, 'AUGMENTED CYCLIC ADVERSARIAL LEARNING FOR LOW RESOURCE DOMAIN ADAPTATION', in ICLR 2019.”
>
> We believe that adding ACAL as a baseline was not necessary because, in this context and for our purposes, ACAL and CycleGAN effectively give us the same result: a function to transform the individual frames of our real-life videos without explicitly modeling the temporal dynamics. Recall that for the CycleGAN baseline in section 4.1 and in table 1, we independently transformed every frame in every video in the real domain. We trained the CycleGAN framework using the individual frames in the real and synthetic training videos. We believe that ACAL would be appropriate for our problem if we had access to very few, albeit labeled, single frames for real-life videos. Instead, we have an abundant number of individual frames in the real domain. In figure 6 in the appendix, the generated thumb tip (top row) and the original thumb (bottom row) are approximately in the same location. This indicates that the generator is producing plausible images and preserving semantic information.

---

> ### Author Response · Authors · 2020-11-15
> **Response to AnonReviewer (Part 1)**
>
> Thank you for taking the time to review our paper and for leaving a detailed review. We address some of the major concerns below:
>
> -> “The model is naive by simply applying the adversarial domain adaptation into the keystroke inference attacks.”
>
> We believe this is an understatement and does not properly characterize the contributions of our work. The bulk of our work is in introducing a method for low resource video domain adaptation, a problem that is not well-explored. We show that by disentangling data into disentangled latent representations (style and content) across domains, we can create style-content pairs that acts as a form of data augmentation. Training with pairs of source and target data has shown to be highly effective in the low resource/few shot domain adaptation scenarios (Motiian et. al 2017, Neurips https://arxiv.org/abs/1711.02536 ), but our method relaxes the constraints of needing pairs of source and target samples to have the same labels. In the “Baseline Results” subsection and table 1, we detail our attempts of applying Adversarial Discriminative Domain Adaptation (ADDA) and a CycleGAN-based method in our setting, but those methods are not suitable for our task of predicting sequences. Since ADDA worked very well for single keypress classification as show in in (Lim et. al 2020), we have added additional text in the related works section and baseline results section to further contextualize our work/results to (Lim et. al 2020), and to further highlight the differences between single key press classification (Lim et. al 2020) and sequence prediction (Ours).
>
> -> “I suspect the paper is more suitable for the security conference, e.g., S&P.”
>
> Our primary contribution is a low resource video domain adaptation framework that leverages supervised disentangled learning. While the application at focus is on keystroke inference attacks, a topic that is prevalent in security venues, our framework is not restricted to keystroke inference attacks. Our method is general enough to be applied to other video problems. Our work would have been more appropriate for security conferences if we were introducing the validity and assessment of a new threat scenario. Our work does not introduce a novel threat scenario. We focus on developing a method such that we can assess previous threat scenarios with an adversary equipped with a deep learning system.

---

### Official Review · AnonReviewer1 · 2020-11-03
**Disentangling style and content for low resource video domain adaptation: a case study on keystroke inference attacks**

**Rating:** 7
**Confidence:** 2

**Review:**

In this paper, the authors focus on keystroke inference attacks in which an attacker leverages machine learning approaches,  In particular, a new framework is proposed for low-resource video domain adaptation using supervised disentangled learning, and another method to assess the threat of keystroke inference attacks by an attacker using a deep learning system, given limited real-life data. The novelty of the approach and its theoretical foundation is appreciated. For a given domain, they decompose the data into real-life style, synthetic style, real-life content, and synthetic content, and then combine them into feature representations from all combinations of style-content pairings across domains to train a model, This allows classify the content of a sample in the style of another domain. Results indicate that training with these pairs to disentangle style and content prevents their model from overfitting to a small real-world training sets, and thereby provides an effective form of data augmentation that prevents overfitting.

The paper is clearly written and well organized, and all the key concepts and motivations are described in enough detail to understand the paper.  However, it is not clear from the outset the amount of limited real-world data should be collected from the target domain. It should also be clarified at the beginning why their data augmentation that prevents models overfitting, and why translated to better security against keystroke inference attacks.

The experimental validation should include an analysis of the impact on the performance of the amount of real-world target domain data, the class imbalance, and capture conditions. The supplementary material provides additional information on the training setup that should be useful to the reader. It however seems like their code is not made available, so there is a concern that the results in this paper would be very difficult for a reader to reproduce.

---

> ### Author Response · Authors · 2020-11-15
> **Response to AnonReviewer1**
>
> We thank the reviewer for their interest in our work. We appreciate the time you took to read the paper and review our work. Below we address any concerns and comments that you have provided.
>
> -> “In this paper, the authors focus on keystroke inference attacks in which an attacker leverages machine learning approaches, In particular, a new framework is proposed for low-resource video domain adaptation using supervised disentangled learning, and another method to assess the threat of keystroke inference attacks by an attacker using a deep learning system, given limited real-life data. The novelty of the approach and its theoretical foundation is appreciated.”
>
> We thank the reviewers for this characterization of our work. This is exactly the message we hoped to convey in our paper.
>
> -> “ it is not clear from the outset the amount of limited real-world data should be collected from the target domain.”
>
> In Table 2, we show performance when using the full real-life training dataset (175 videos) as well as a truncated version of the real-life training dataset (100).
>
> -> “It should also be clarified at the beginning why their data augmentation that prevents models overfitting”
>
> When dealing with scarce data in the target domain, training on pairs of source and target data samples have been proven to be effective in dealing with overfitting ( Motiian et al 2017, ICCV https://arxiv.org/abs/1709.10190, Motiian et. al 2017, Neurips https://arxiv.org/abs/1711.02536,   Wang et. al 2019, CVPR https://arxiv.org/abs/1903.09372  ). These pairing mechanisms, as well as ours, attempts to augment the target dataset size by the order of the source domain. We have added additional text in the introduction to highlight that this training recipe has been shown as an effective form of data augmentation, and to highlight the novelty in our pairing/grouping strategy.
>
> -> “The experimental validation should include an analysis of the impact on the performance of the amount of real-world target domain data, the class imbalance, and capture conditions”
>
> In Table 2, we show performance when using the full real-life training dataset (175 videos) as well as a truncated version of the real-life training dataset (100). We have added further details in the datasets subsection in our experiments section to further detail the capture conditions. Unfortunately, we did not keep track of additional data to indicate different capture conditions so it is not possible to give an analysis as to how outdoor vs indoor capture scenarios, for example, effect performance.
>
> -> “The supplementary material provides additional information on the training setup that should be useful to the reader. It however seems like their code is not made available, so there is a concern that the results in this paper would be very difficult for a reader to reproduce.”
>
> We are glad that you found the training details sufficient for future reproducibility. We plan on releasing the code + dataset (real and synthetic) to the public after the review process.
>
>
> Again, we thank you for taking the time to review our work. We are grateful for the positive review and have incorporated your feedback into our paper.

---

### Comment · Area_Chair1 · 2020-11-20
**Please, check rebuttals and provide further feedback to authors if needed**

Dear Reviewers and Authors,
Thanks for starting the discussion.

Reviewers: please, check the rebuttals provided by the authors, verify if they replied properly and you are satisfied.
Possibly, give further feedback or make questions, only if needed and important for your final evaluation.
Please, be accurate and precise in your further requests, so that authors can understand and reply properly and focused.

Authors: check if there are further clarifications needed by the Reviewers.
Please, be focused in your final answers and avoid to ask questions to Reviewers, if not absolutely necessary.

For all: please, I would avoid a long chat-like discussion, a couple of iterations are affordable on a few specific points to be clarified, but no more.

Thanks and best regards

AC

---

### Decision · Program_Chairs · 2021-01-07
**Final Decision**

**Decision:**

Reject

**Comment:**

The paper focuses proposes a new framework for low-resource video domain adaptation leveraging synthetic data with supervised disentangled learning for tackling keystroke inference attacks.

The paper received contrasting reviews, 2 positive and 2 negative, and the overall confidence of the reviewers is not so high.
Overall, it is recognized that the work has some merit, but also some problems, which the rebuttal has not fully fixed, and I mainly refer to R2, R3 and R4 remarks.

The first issue is the level of novelty, which is not much high as compared to the former work of (Moiitan et al. 2017). Besides, the questions raised by some reviewers, also discussed in the rebuttal, also denote a certain lack of clarity, despite the paper is considered well organised in general.

The other main issue regards the experimental evaluation. To start with, the application addressed is very specific and it is not clear how this approach can be extended to other problems too, since no evidence is provided in this sense. The reported comparative analysis wrt baselines are in fact quite "simple" (e.g., ADDA is a work dated back to 2017, so as CycleGAN). Moreover, although the considered dataset is the only one in this scenario, its significance is a bit limited since only 3 subjects were considered, and this likely raised the comment of one reviewer questioning if this paper was not better suited to an application-oriented, security conference. A discussion for the setting of the lambda parameters is also missing.

Overall, given the above issues, I consider the paper not yet ready for publication in ICLR 2021.